



# PROMICE-2022 Ice Mask: A high-resolution outline of the Greenland Ice Sheet from August 2022

Gregor Luetzenburg[1*], Niels J. Korsgaard[1*], Anna K. Deichmann[1], Tobias Socher[1], Karin Gleie[1], Thomas Scharffenberger[1], Rasmus P. Meyer[2], Dominik Fahrner[1], Eva B. Nielsen[1], Penelope How[1], Anders A. Bjørk[1,2], Kristian K. Kjeldsen[1], Andreas P. Ahlstrøm[1], Robert S. Fausto[1]

*These authors contributed equally to this work.

[1]Department of Glaciology and Climate, Geological Survey of Denmark and Greenland (GEUS), Copenhagen, Denmark

[2]Department of Geosciences and Natural Resource Management, University of Copenhagen, Copenhagen, Denmark

*Correspondence to:* Gregor Luetzenburg (grelu@geus.dk)

**Abstract.** The Greenland Ice Sheet (GrIS) is losing mass at an accelerating rate in response to climate change. Its geometry responds to these changes over annual to decadal timescales, therefore making accurate and up-to-date mapping of its extent essential for monitoring ice loss, assessing mass balance, and improving climate and glaciological models. Ice margin outlines serve as critical boundary conditions for different types of modelling exercises, hydrological studies, and assessments of ice sheet dynamics. Here, we present the PROMICE-2022 Ice Mask, a high-resolution outline of the contiguous ice masses of the GrIS and the nunataks in its interior as of late August 2022. The dataset was derived from a true-colour Sentinel-2 mosaic at 10 m spatial resolution, generated using the Sentinel Hub Cloud Processing API to select the most recent valid pixels from August 2022. The mapping process was performed manually and supplemented with high-resolution mosaics from Sentinel-2 and SPOT 6/7 provided by the Danish Agency for Climate Data (KDS), along with recent topographic vector data. We mapped the geodesic perimeter length of the GrIS to 53,060 km and its glacierized area to 1,725,648 km² with 19,130 nunataks in its interior. The PROMICE-2022 ice mask captures the GrIS margin with an absolute horizontal accuracy better than 20 m. Its quality and consistency make it well suited for applications in ice sheet modelling, hydrology, glacial geomorphology, and long-term monitoring of ice margin change. The complete dataset is freely available for download at https://doi.org/10.22008/FK2/O8CLRE (Luetzenburg et al., 2025).



## 1 Introduction

The Greenland Ice Sheet (GrIS), the second-largest ice expanse on Earth, has been losing mass at an accelerating rate (Imbie, 2020; Mouginot et al., 2019), contributing at least $274 \pm 68$ mm to global sea-level rise from 2000 to 2019 (Box et al., 2022). This trend is expected to continue in the future (Aschwanden et al., 2019; Goelzer et al., 2020; Pattyn et al., 2018; Choi et al., 2021). The geometry of the GrIS responds to fluctuations in atmospheric and oceanic conditions over annual to decadal time scales (Kjeldsen et al., 2015), and the current rapid shrinkage of the ice sheet and the detached ice

bodies  have immediate and, at times, severe consequences for human populations both locally and globally, for instance in terms of bedrock instability due to glacial retreat (Greene et al., 2024; Svennevig et al., 2024). Therefore, accurate mapping of the GrIS extent is crucial for monitoring ice loss, assessing mass balance, and refining climate models (Howat et al., 2014). Ice sheet outlines provide key boundary conditions for glaciological modelling, hydrology, and ice margin studies.

Due to its vast extent, remote location, sparse population, and minimal infrastructure, it is impossible to map the outline of the GrIS and the nunataks in its entirety in the field. Aerial and satellite images offer an opportunity to survey the GrIS remotely with a medium-to-high level of detail and decent ground control (Paul et al., 2017a; Paul et al., 2015). The Sentinel-2 satellite provides freely available, true-colour images with a high spatial resolution (~10 m) and frequent revisit times, enabling high-precision delineation of the ice margin. However, mapping glacier ice from space is challenging in

Greenland due to the large latitudinal extent, debris cover, seasonal snowfall, and classification uncertainties (Rastner et al., 2012; Paul et al., 2017b). Automated methods for glacier ice delineation across large scales are improving, however, accurate delineation, especially in debris-rich areas continues to require refinement (Paul et al., 2016; Roberts-Pierel et al., 2022; Maslov et al., 2025).

Several datasets provide mapped outlines of the GrIS, each with unique strengths and limitations. The widely used GIMP

(Greenland Ice Mapping Project) ice mask offers some of the highest-resolution outline (15 x 15 m), making it widely used in ice margin tracking and mass balance studies (Howat et al., 2014). BedMachine v5, which includes an ice and ocean mask from GIMP, uniquely combines ice thickness, ice velocity, and subglacial topography data; however, it features a lower spatial resolution ($150 \times 150$ m) for defining ice margins (Morlighem et al., 2017). Earlier datasets, such as Rastner et al. (2012) (hereafter referred to as RGI) remain essential for studying Greenland's peripheral glaciers, with

manually corrected outlines derived from satellite imagery and digital elevation models. However, a limitation, in particular for older datasets, is a clear definition of the origin of the input data and a coherent timestamp associated with this; for instance the ESA Climate Change Initiative ice mask (Esa, CCI) is based on  Landsat-7 imagery input data spanning 1994 to 2009, with core data acquired during 1999-2004, and moreover, in  far northern Greenland, it uses the GIMP ice mask. Stereo-photogrammetric imagery from a large air-borne campaign during 1978-1987 provides the input

to the first ice mask of the Programme for Monitoring of the Greenland Ice Sheet (hereafter referred to as PROMICE-1987 ice mask), but in the northwest the ice mask mapping is produced using older aerial photographs than the 1978–1987 campaigns (Citterio and Ahlstrøm, 2013). During the past decade, more data has become available, making temporal coverage less of an issue. Additionally, operational datasets such as the Åbent Land Grønland (GL50), provided by the Danish Agency for Climate Data, serve primarily as a topographic reference, with limited application in dynamic ice sheet

studies (Kds, 2025b), leaving it less desirable to rely on for cryospheric investigations.

To reconcile mass balance estimates, recent efforts have standardized ice sheet masks and assessment periods, ensuring that any discrepancies come from methodological differences rather than spatial or temporal inconsistencies (Imbie, 2020; Sasgen et al., 2012; Shepherd et al., 2012; Vernon et al., 2013). While these efforts are a fundamental necessity in cryospheric research, the resulting mass balance estimates for recent years may be biased by using ice masks that do not track recent years' ice margin migration (Kjeldsen et al., 2020). Here, we present the PROMICE-2022 ice mask, an expert-based, manually mapped vector outline of the contiguous ice masses of the Greenland Ice Sheet, derived from Sentinel-2 satellite imagery acquired in August 2022. Mapping was performed at a scale of around 1:25,000, after which quality assessment was performed independently of the mapping operator, before finally being merged into one coherent dataset.

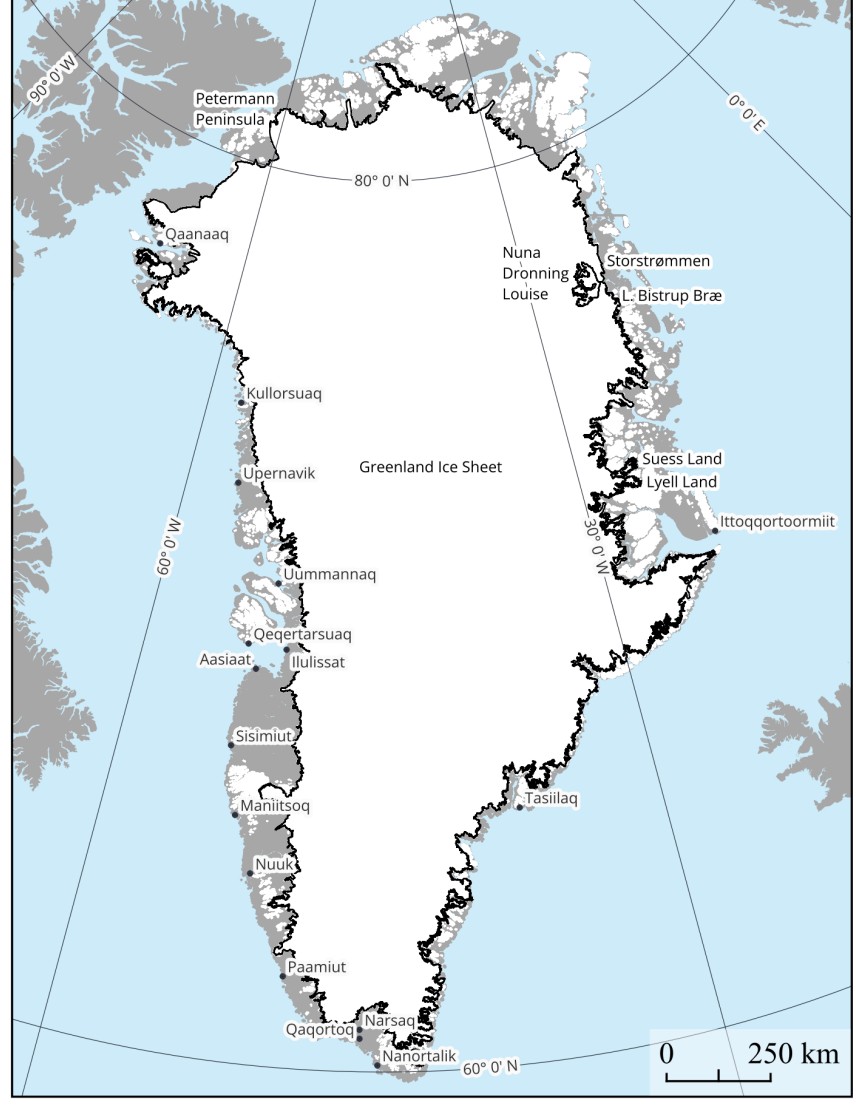

**Figure 1:** Overview map of Greenland showing populated places and relevant geographic names. Ice-free land areas are shaded in grey, and ice-covered regions are shown in white. The outline of the PROMICE-2022 ice mask is indicated by a solid black line. Base maps for plotting are from QGreenland v3.0 (Moon et al., 2023).

## 2 Data sources

### 2.1 August 2022 Sentinel-2 mosaic

For the GrIS margin and the nunataks in its interior, a multi-band, true-colour mosaic was compiled from Sentinel-2 data using the Sentinel Hub Processing API, a cloud-based platform for satellite data processing (Sinergise Solutions, 2025). An area-of-interest mask was applied to control the spatial extent of the mosaic, encompassing the GrIS margin, nunataks, and the surrounding area. This process generated a pixel-by-pixel, multi-band mosaic using the most recent August 2022 Sentinel-2A data. As part of the API request, the data was filtered to exclude poor-quality data (e.g., cloud-covered areas),
ensuring that only valid pixels were included within the specified time frame.

In addition to the Sentinel-2 mosaic, we generated a vector polygon layer indicating the Day of Year (DOY) of image acquisition for each pixel within the mosaic. Approximately 93 % of the area was acquired between DOY 213–243, corresponding to August 2022, while less than 2 % of the area was acquired between DOY 182–212 (July 2022). Fewer than 5 % has no available DOY information. In some areas, the GrIS outline was mapped beyond the predefined area of
interest. For these cases, we manually downloaded additional Sentinel-2 scenes from August 2022 via the Copernicus Browser to cover the required regions.

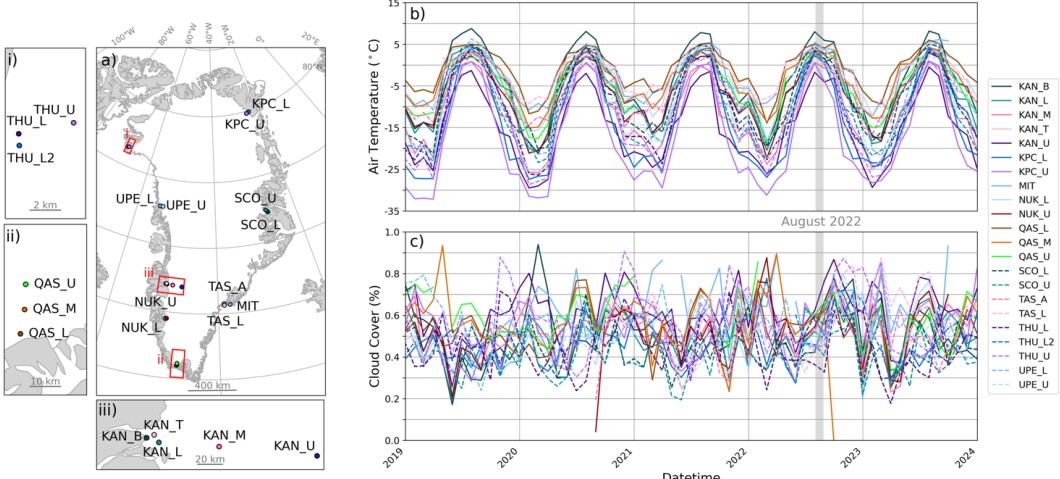

**Figure 2:** Assessment of optimal conditions for ice margin delineation, based on in situ temperature and cloud cover observations from the PROMICE automated weather station network (Fausto et al., 2021). Monthly averaged observations are presented from weather
stations positioned at the periphery of the GrIS (a), collating temperature (b) and cloud cover (c) measurements between 2019 and 2024. August 2022 is highlighted on the graphs (grey) as a suitable period, with high temperatures between –4 °C (at KPC_U and KAN_U) and 7 °C (KAN_B), and cloud cover ranging from 33 % (THU_L) to 70 % (KAN_U).



Since most of the GrIS terminates on land, the absence of snow in late August makes this the ideal time of the year for acquiring snow-free remote sensing data (Fig. 2). During this period, the ablation zones of the GrIS margin, along with peripheral glaciers and ice caps, are largely snow-free, simplifying the delineation of glacier boundaries between ice and land. For example, in North Greenland, snow cover typically disappears in July and returns in September (Cappelen and Drost Jensen, 2021). This seasonal pattern determines the data acquisition time frame. While marine-terminating glaciers generally reach their maximum retreat between October and early November before beginning their seasonal readvance (Black and Joughin, 2023), these glaciers represent only a small fraction of the GrIS perimeter. Therefore, the need for optimal mapping conditions, achieved with late August snow-free imagery, took precedence over capturing the annual maximum retreat of calving fronts.

**2.2 KDS Sentinel-2**

The Danish Agency for Climate Data (KDS; Danish: Klimadatastyrelsen) produced a series of Sentinel-2 mosaics (~10 m resolution) for the consecutive years 2019–2022, which are available as Web Map Service (WMS) and can also be downloaded via File Transfer Protocol (https://dataforsyningen.dk/data/4783). The 2022 mosaic was compiled from Sentinel-2 data spanning May 28 to September 12, although in practice, almost all tiles are from the months of June–August, with August tiles making up 68 % of the mosaic. The Sentinel-2 images used in the mosaic are orthorectified using either the GLOBE (1 km) or SRTM (30 m) elevation models (Kds, 2025a).

**2.3 KDS SPOT 6/7**

KDS is also providing a mosaic of high-resolution SPOT6/7 satellite images (~1.6 m resolution) that can be accessed via the same WMS link as the Sentinel-2 images (Kds, 2025a). The SPOT 6/7 satellite images were primarily recorded in 2020 (75 %), supplemented by images from 2016 (25 %) and a few acquisitions from 2013 and 2014. While much of Greenland is covered by the SPOT 6/7 mosaic, there are areas that remain uncovered. The WorldDEM4Ortho (Airbus, 2018) was used for orthorectification.

**2.4 KDS Åbent Land Grønland (GL50)**

Åbent Land Grønland (GL50) is a 1:50,000 scale topographical map of Greenland, created using 0.5 m high-resolution commercial satellite imagery as the source data (Kds, 2025b). It covers both the GrIS and the glaciers and ice caps (GICs). The SPOT 6/7 images and the 1:50,000 scale topographical map of Greenland can be easily accessed via QGreenland v3.0 (Moon et al., 2023). The source data for the entire vector dataset is primarily from the summer months between 2017 and 2021. Specifically, for glacier and snowfield coverage, the acquisition date range spans from July 2013 to October 2020. Currently, 58 % of the glacier outlines have unspecified acquisition dates in the data set. For outlines with assigned source dates, most fall within the period from 2015 to 2020 (41 %). We extracted the surface classes "glacier_s" and "snowicefield_s" from the full dataset, and these were subsequently merged and converted those to vector lines. These lines were then generalized using the Douglas–Peucker (1973) algorithm with a maximum tolerance of 5 meters (equivalent to half a pixel). This simplification reduced the number of vertices by approximately 70 %, introduced a negligible maximum error of 5 m and a mean error of 1.3 m, and significantly decreased the amount of manual editing required during subsequent revisions.



## 3 Methods

### 3.1 GrIS outline and nunatak mapping

The PROMICE-2022 ice mask was manually mapped in a custom QGreenland v3.0 QGIS environment (Moon et al., 2023), by tracing and delineating the GrIS margin directly from the August 2022 Sentinel-2 mosaic. In areas where the Sentinel-2 mosaic was not available or not of sufficient quality, we used the Sentinel-2 from 2022 provided by KDS. Rarely, as a third alternative, we consulted the Spot 6/7 images also provided by KDS to delineate the GrIS outline where the available Sentinel-2 images from 2022 rendered unusable. A common practice while delineating the outline was to

use the high-resolution Spot 6/7 images to differentiate between snow, ice or debris and subsequently map the exact position of the GrIS margin from the Sentinel-2 mosaic from August 2022. In some cases, we used the GL50 topographic map from KDS as a reference for revision and editing where appropriate. In areas with unfavourable lightning conditions, often due to steep topography, we used a multidirectional hillshade of the Arctic DEM (Porter et al., 2024) with 10 m

resolution to guide our PROMICE-2022 ice mask delineation. For the nunatak delineation, we used a combination of manual mapping and editing the nunataks provided in the GL50 dataset. Here, we also used the Sentinel-2 mosaic from 2022 as the primary reference and only occasionally the Sentinel-2 mosaic from 2022 or the SPOT 6/7 images (Fig. 3).

    The 10 m resolution of the Sentinel-2 and 1.6 m resolution of the SPOT 6/7 satellite images are suitable for cartographic mapping to scales of 1:20,000 and 1: 3,200, respectively (Tobler, 1987). For the delineation of the GrIS margin and

nunataks, we used a variable mapping scale adapted to the level of detail required in different regions. Areas with complex topography or intricate transitions between ice, snow, and sediment were mapped at finer scales than more uniform sections of the ice margin. Throughout the mapping process, we aimed for a minimum mappable feature (MMF) of approximately 1 mm on screen, which corresponds to a mapping scale of about 1:25,000. This scale aligns well with the 10 m spatial resolution of the Sentinel-2 imagery used, translating to an MMF of approximately 25 m on the ground

(Drusch et al., 2012).

    The delineation of the GrIS divides the surface of Greenland into two classes: the PROMICE-2022 ice mask, and everything else. The ice mask includes floating ice tongues and thus was not limited to grounded ice only. A topological definition was applied, consistent with the connectivity concept used in Rastner et al. (2012) and for the GIMP ice mask (Howat et al., 2014). This definition includes all contiguous ice as part of the GrIS, encompassing the ice sheet "proper",

as well as GICs that are connected to it. It also includes debris-covered ice (e.g., lateral and medial moraines), stagnant or dead ice, snowfields, snowdrift glaciers, and supraglacial lakes. Many GICs peripheral to the GrIS are considered connected either through the accumulation area, via a local glacier or ice cap that spans the drainage divide, or where they coalesce with the GrIS in the ablation area. Following connectivity level 1 (CL1) defined by Rastner et al. (2012) we separately delineate GICs that are clearly divided by drainage in the accumulation region and are not connected in the

ablation region. In contrast, GICs that coalesce with the GrIS in the ablation region (CL2) are not delineated separately. Additionally, we adapted the glacier catchments provided by Mouginot et al. (2019) to the PROMICE-2022 ice mask.

    One notable distinction between the PROMICE-2022 ice mask and the GIMP dataset is the treatment of ice-dammed lakes. While the GIMP mask includes ice-dammed lakes, the PROMICE-2022 ice mask does not. Ice-dammed lakes are excluded because the presence and volume of liquid water in these lakes can fluctuate due to variability in meltwater



runoff and periodic drainage events (Dømgaard et al., 2024). The glacier outlines of Rastner et al. (2012) similarly excluded ice-dammed lakes, identifying frozen lakes using hillshaded DEMs and multi-temporal satellite imagery. In our dataset, ice-dammed lakes were identifiable in 10 m Sentinel-2 and 1.6 m SPOT 6/7 true-colour mosaics regardless of whether they were frozen or not. Additionally, lakes in proximity to the ice sheet were detected using vector data from the GL50 topographical dataset (Kds, 2025b). In contrast to the GL50 ice and snow delineation, we included debris

covered ice as part of the PROMICE-2022 ice mask which is especially notable at large valley and outlet glaciers with prominent medial moraines.

Snow presented a particular challenge in the delineation of the PROMICE-2022 ice mask. The extent of snow cover is highly variable, influenced not only by seasonal changes but also by occasional snowfall shortly before satellite image acquisition. To minimize the impact of snow cover, Sentinel-2 data from as late in August as possible were used for the

mosaic. During manual editing and tracing, the KDS Sentinel-2 and SPOT 6/7 mosaics were also utilized to minimize the extent of seasonal and occasional snow cover.

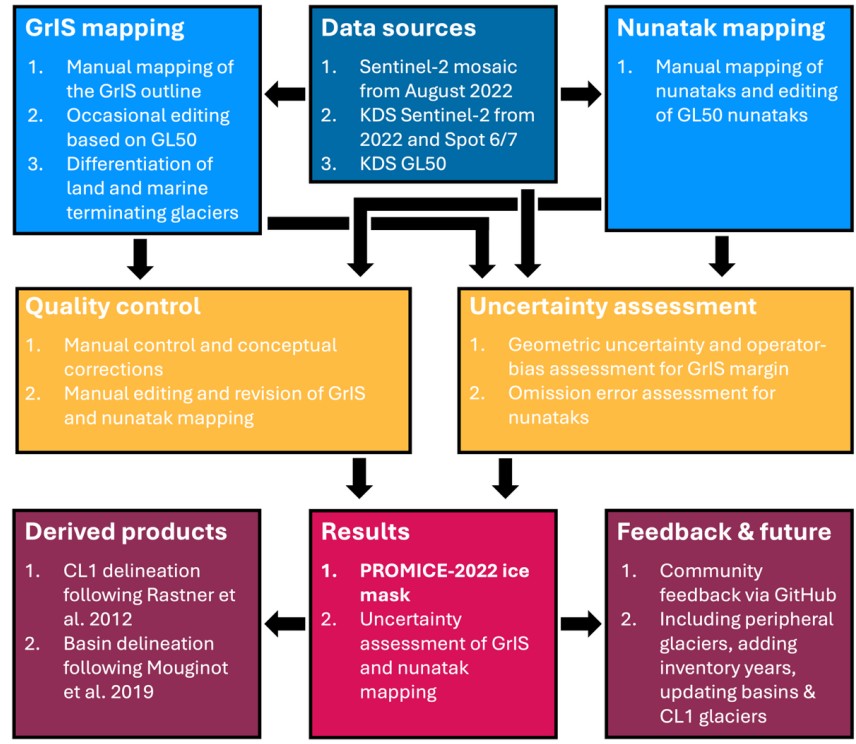

**Figure 3:** A visualisation of the processing workflow for the generation of the PROMICE-2022 ice mask.





### 3.2 Quality control and uncertainty assessment

The PROMICE-2022 ice mask and the nunataks within it were divided into several zones, each mapped by different experts following the outlined procedure. To ensure consistency across all of Greenland, each zone was independently reviewed, edited, and refined by a second expert. Quality control included checks for both completeness and correctness, including the verification of conceptual issues such as the connectivity of GICs to the GrIS. Corrections were either made directly by the reviewing expert or returned to the original mapper for revision, followed by a second round of review when necessary.

The absolute accuracy, defined as the measure of how close the vectorized outline is to its true location, can be considered a combination of two main sources of error: the geometric accuracy of the satellite imagery and the operator error associated with outlining the GrIS. Geometric and operator uncertainties were assessed for the PROMICE-2022 ice mask outline, and we assume these uncertainties are comparable for the nunatak mapping. Total uncertainty is expressed as the square root of the sum of the variances of the geometric uncertainty and the operator bias (RSS) (Jcgm, 2008). Given the large number and variable spatial extent of the nunataks, we additionally assessed the omission error to quantify and reduce the likelihood of overlooked features during the initial mapping and quality control process.

### 3.2.1 Geometric uncertainty

Geometric uncertainty describes the positional accuracy of a feature, and it is influenced by uncertainties in the satellite's orbit and sensor pose (Kääb et al., 2016). An additional source of uncertainty is introduced by the digital elevation models (DEMs) used for ortho-rectifying satellite imagery. Sentinel-2 data in Greenland are ortho-rectified using either the GLOBE (1 km) or SRTM (30 m) elevation models (Kds, 2025a). Kääb et al. (2016) found that the most significant geometric issues arise from vertical errors in DEMs, which propagate into lateral offsets of several pixels. Greenland's dynamic landscape, including the changing ice elevation at the margins and outlet glaciers, exacerbates these errors leading to geometric accuracies ranging between 1-2 pixels (10-20 m) up to 10 pixels (100 m). The use of a static DEM introduces inaccuracies not only when comparing the ice elevation with other georeferenced data but also between data from different relative orbits of Sentinel-2 (Kääb et al., 2016). Here, we cannot directly assess the errors resulting from the DEMs and our evaluation will focus on assessing the relative geometric uncertainty.

We used the Aero orthophoto as reference to calculate the relative geometric uncertainty of the Sentinel-2 mosaic. The Aero orthophoto was created from approximately 3500 aerial photographs recorded 1978-1987, covering the entire ice-free land margin and the ice sheet periphery of Greenland (Korsgaard et al., 2016). With field surveyed geodetic ground control points and a two-meter resolution, the Aero orthophoto is the most accurate, high-resolution orthophotograph available all-around Greenland. We assessed the geometric uncertainty by mapping 600 point features on clearly visible landmarks in close vicinity to the GrIS outline on the Sentinel-2 mosaic from August 2022 and the Aero orthophoto. The 600 mapped features are distributed all around Greenland, and the distances between the mapped points at the Aero orthophoto and the Sentinel-2 mosaic are a relative assessment of the geometric uncertainty.



### 3.2.2 Operator bias

Operator error arises from subjective decisions made during the manual interpretation and editing of the vector layer superimposed on the true-colour mosaic. To quantify this operator bias, we created 304 transect lines, each intersecting the GrIS outline once. These lines varied in length (from approximately 2.7 km to 74 km) and spacing. An independent expert, using the same datasets and applying the same mapping criteria as the original mappers, but without access to their results, placed one point along each line at what they perceived to be the margin of the GrIS. The distance between

these independently mapped points and the corresponding intersections with the GrIS-2022 outline serves as a measure of operator bias.

However, we observed that mapping the GrIS margin as a continuous line leads to different decisions at a larger spatial scale than mapping isolated points along transects. This difference arises because decisions about margin continuity, especially in complex or ambiguous areas, are influenced by the broader spatial context. Therefore, we limited our

assessment of operator bias to a local scale and excluded 45 measurements where distances exceeded 100 m, resulting in a final sample of 259 points. Broader decisions, such as the connectivity of local ice caps to the main ice sheet were made collaboratively among experts and are not included in the operator bias assessment.

### 3.2.3 Omission error

To further validate the nunatak mapping, another expert independently mapped nunataks within a randomly selected

subsample area. This validation aimed to assess the completeness of the initial mapping (Luetzenburg et al., 2022). The entire area of investigation was divided into 286 tiles, each measuring $50 \times 50$ km. From these, 23 tiles were randomly selected, forming a subsample with a confidence level of approximately 90 % and a margin of error of ±16.5 %. The independent mapper used the same datasets and followed the same mapping criteria as the initial mappers but was blind to their results in the selected subset. During quality control, the independent expert digitized one point per nunatak. A

match was recorded when a point mapped by the third expert fell within a nunatak polygon previously identified by the initial mappers. Based on this comparison, additional nunataks identified by the independent expert were incorporated into the final inventory when they were deemed valid by the initial mappers. However, in some cases, the initial mappers disagreed with the independent expert's interpretation, and not all newly mapped features were retained in the final database.

### 240 3.3 The PROMICE-2022 ice mask dataset format and structure

The mapped and quality-controlled PROMICE-2022 ice mask margin and nunatak features were checked and corrected for dangling nodes, duplicate features, and other invalid geometries. The vector lines representing the margin and each individual nunatak were then dissolved and simplified by distance, using a tolerance of 5 m, equivalent to half a pixel (Douglas and Peucker, 1973). Mean error of the simplification is 1.3 m. Subsequently, the simplified lines were densified,

ensuring a maximum spacing of 100 m between vertices. Table 1 summarizes the different file configurations of the PROMICE-2022 ice mass that are being provided. The definitions of the attribute fields associated with the corresponding data products are provided in Table 2. The data are distributed as line and polygon vector features as well as raster features in the GeoPackage format (.gpkg), with coordinates provided in the WGS NSIDC Sea Ice Polar Stereographic North (EPSG:3413) projected coordinate system.



**Table 1.** PROMICE-2022 ice mask file configurations provided at the GEUS Dataverse

| No | Name | Format | Description |
|---|---|---|---|
| 00 | README-PROMICE-2022-IceMask | md | The dataset readme file |
| 01 | PROMICE-2022-IceMask-line | gpkg | Outline of the Greenland Ice Sheet from August 2022, provided as line vector feature |
| 02 | PROMICE-2022-IceMask-polygon | gpkg | Outline of the Greenland Ice Sheet from August 2022, provided as polygon vector feature |
| 03 | PROMICE-2022-Nunatak-line | gpkg | Outlines of the nunataks within the Greenland Ice Sheet from August 2022, provided as line vector features |
| 04 | PROMICE-2022-Nunatak-polygon | gpkg | Outlines of the nunataks within the Greenland Ice sheet from August 2022, provided as polygon vector features |
| 05 | PROMICE-2022-IceMask-Nunatak-line | gpkg | Outlines of the Greenland Ice Sheet and the nunataks within from August 2022, provided as line vector features |
| 06 | PROMICE-2022-IceMask-Nunatak-polygon | gpkg | Outline of the Greenland Ice Sheet from August 2022 with the nunataks in its interior cut out, provided as a polygon vector feature |
| 07 | PROMICE-2022-IceMask-CL1-polygon | gpkg | Outline of the Greenland Ice Sheet from August 2022 with the nunataks in its interior cut out, and glaciers with connectivity level CL1 delineated following Rastner et al. 2012, provided as a polygon vector feature |
| 08 | PROMICE-2022-IceMask-basins-polygon | gpkg | Ice mask of the Greenland Ice Sheet from August 2022, divided into drainage basins following Mouginot and Rignot 2019, provided as a polygon vector feature with an associated layer definition file (.qlr) |
| 09 | PROMICE-2022-IceMask-Nunatak-basins-polygon | gpkg | Ice mask of the Greenland Ice Sheet from August 2022 with the nunataks in its interior cut out, divided into drainage basins following Mouginot and Rignot 2019, provided as a polygon vector feature with an associated layer definition file (.qlr) |
| 10 | PROMICE-2022-IceMask-raster-150m | gpkg | Ice Mask of the Greenland Ice Sheet from August 2022 with the nunataks in its interior cut out, provided as raster file with a cell size of 150x150m sing the same grid as BedMachine Greenland following Morlighem et al. 2017 |
| 11 | PROMICE-2022-DOY-polygon | gpkg | Polygon vector feature of the Sentinel-2 mosaic extent, annotated with the DOY (day of year) in 2022 on which the corresponding Sentinel-2 imagery was acquired and used for the delineation of the PROMICE-2022 ice mask |



| 12 | PROMICE-2022-DOY-line | gpkg | The PROMICE-2022 Ice Mask and Nunatak lines attributed with the DOY from the PROMICE-2022 DOY polygons |
|---|---|---|---|

**Table 2:** Attributes included with PROMICE-2022 ice mask configurations

| Variable name | Description | Format |
|---|---|---|
| id | Identifying number for each feature | Integer |
| type | Ice sheet outline or nunatak (outline, nunatak) | String |
| termini | Land or marine terminating ice sheet margin (land, marine) | String |
| MTG_ID | Marine terminating glacier identifying number | Integer |
| area_planar_sqkm | Planar areal extent of polygons in square km | Float |
| area_geodesic_sqkm | Geodesic (WGS 84) areal extent of polygons in square km | Float |
| length_planar_km | Planar length of line or polygon perimeter in km | Float |
| length_geodesic_km | Geodesic (WGS 84) length of line or polygon perimeter in km | Float |
| connectivity | Connectivity level as defined by Rastner et al. 2012 (CL1) | String |
| subregion | Region as defined by Mouginot and Rignot (2019) (NW, NO, NE, CE, SE, SW, CW) | String |
| DOY | Day of year in 2022 on which the corresponding Sentinel-2 imagery was acquired | Integer |

## 4 Results

### 4.1 Ice mask and nunataks

The final PROMICE-2022 ice mask vector product is a synchronous polygon layer representing the extent of all contiguous glacierized areas of the Greenland Ice Sheet (Fig. 4). Based on this dataset, we calculate the total, glacierized geodesic area of the GrIS to be 1,725,648 (+/- 1,061) km², excluding the area occupied by the nunataks. The geodesic perimeter length of the PROMICE-2022 ice mask margin is 53,060 km, with 2,386 km marine terminating glacier fronts divided into 841 individual glaciers, and 50,674 km land terminating ice margin. We mapped 19,130 nunataks within the PROMICE-2022 ice mask with a total, projected, area of 17,916 (+/- 1,674) km² (Fig. 5). Figure 4a and 5a show the ice sheet outline and nunataks of the PROMICE-2022 ice mask, respectively, while Fig. 4b–g and 5b–g illustrate mapping decisions related to the ice mask and nunatak delineation, overlaid on the August 2022 Sentinel-2 mosaic.

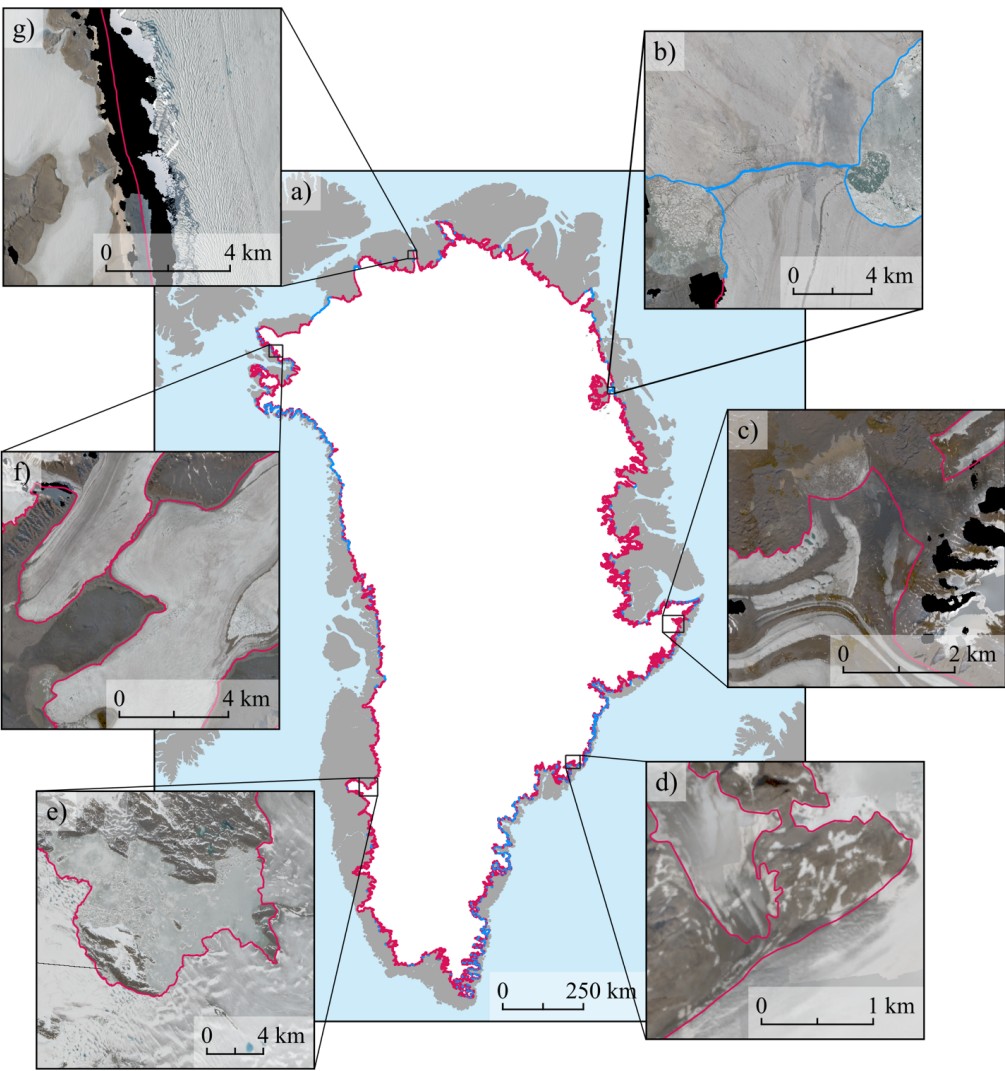

**Figure 4:** (a) The PROMICE-2022 ice mask with land terminating sections shown in red and marine terminating sections in blue. Panels b-g illustrate mapping decisions in the PROMICE-2022 ice mask overlaid on the August-2022 Sentinel-2 mosaic (Copernicus Sentinel data (2022), processed via Sentinel Hub): (b) the confluence of two marine-terminating glacier fronts; (c) a debris-covered margin with small ice-free patches and unclear delineation; (d) a clearly defined ice margin in eastern Greenland, highlighting the subtle criteria used to the ice sheet boundary versus disconnected ice patches; (e) a frozen ice-marginal lake excluded from the GrIS mask; (f) an area where glacier connectivity is ambiguous due to narrow ice tongues; and (g) a region affected by poor or missing image data.



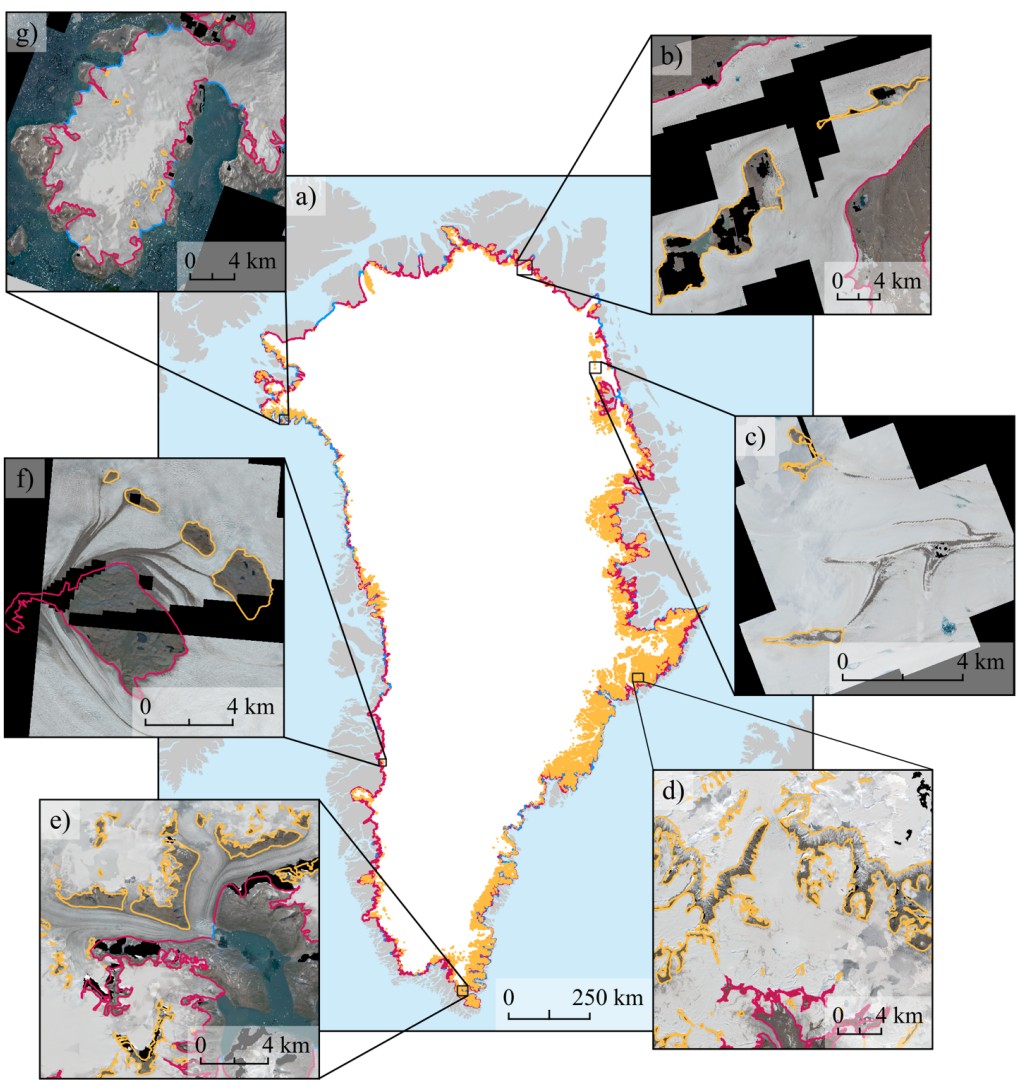

**Figure 5:** (a) The PROMICE-2022 ice mask with land-terminating sections shown in red, marine-terminating sections in blue, and nunataks in yellow. Panels (b–g) illustrate mapping decisions related to nunatak delineation, overlaid on the August 2022 Sentinel-2 mosaic (Copernicus Sentinel data (2022), processed via Sentinel Hub): (b) nunataks and ice-marginal lakes in a region partially affected by missing data; (c) nunataks adjacent to elongated, debris-covered ice features and small supraglacial lakes; (d) complex, branched nunatak geometries in eastern Greenland; (e) nunataks situated between debris-covered, converging ice streams in southern Greenland; (f) nunataks near the ice margin in western Greenland, with sediment tails aligned with ice flow; and (g) nunataks atop a local ice cap in northwestern Greenland with marine-terminating outlet glaciers.



### 4.2 Quality control and uncertainties

The median relative geometric uncertainty of the Sentinel-2 mosaic is 11.9 m (mean: 12.7 m) and the median operator bias is 11.1 m (mean: 17.6 m, Fig. 6a). The combined RSS of geometric uncertainty and operator bias is 19.5 m. Geometric uncertainty and operator bias are spatially consistent across Greenland. The assessment of the completeness of the nunatak

mapping showed that the independent mapper identified 1,165 nunataks within the selected areas, of which 1,141 matched features in the PROMICE-2022 nunatak mapping, and 24 were not present in the original dataset (Fig. 6b). This corresponds to an omission error of approximately 1 % in the PROMICE-2022 nunatak delineation. However, the independent mapper missed to identify 1,178 nunataks originally mapped in the PROMICE-2022 dataset, suggesting differences in interpretation or detection threshold between the two mapping efforts. Thus, the total number of nunataks

in the selected area is 2,343.

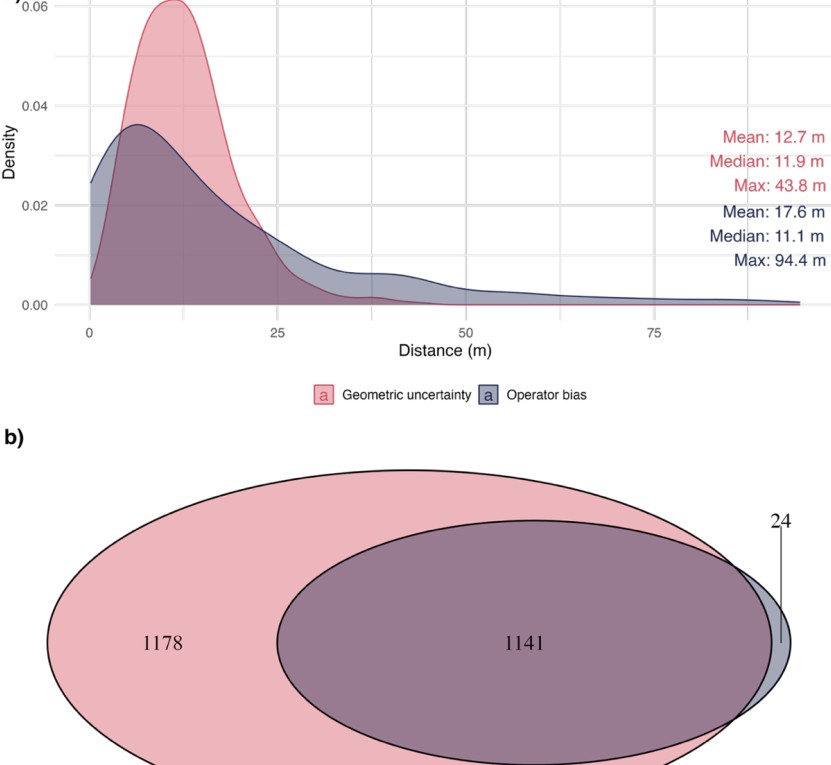

**Figure 6:** Uncertainties of the PROMICE-2022 ice mask. (a) Density distributions of geometric uncertainty and operator bias expressed as Euclidean distances (in meters) between mapped features. Geometric uncertainty reflects discrepancies due to image resolution and geolocation, while operator bias captures variation between independent interpretations. (b) Venn diagram illustrating omission error

based on comparison with an independent mapping: 1,141 nunataks were identified by both mappers, 1,178 were unique to the PROMICE-2022 nunatak mapping, and 24 were omitted and only identified by the independent mapper.





## 5 Discussion

### 5.1 Uncertainties and biases

Delineating the PROMICE-2022 ice mask margin and nunataks involves uncertainties from image resolution, mapping
scale, and operator interpretation. Despite efforts to improve precision, like adjusting scale to local complexity and
maintaining a ~25 m MMF, some limitations persist. The use of Sentinel-2 imagery, with a 10 m spatial resolution,
constrains the level of detail that can be confidently interpreted, particularly in areas where ice, snow, and sediment
boundaries are diffuse or seasonally variable. Although a mapping scale of 1:25,000 is theoretically compatible with the
resolution of the imagery, the practical limit of delineation is influenced by image quality (e.g., contrast, shadowing) and
surface conditions at the time of acquisition. The chosen MMF of ~1 mm on screen is appropriate for most use cases but
may not capture very narrow features that are smaller than 25 m in width. The relative consistency of the uncertainties
across Greenland helps ensure internal coherence within the dataset but does not eliminate localized misalignments.

Regions with complex terrain or unclear ice boundaries require finer-scale mapping, which improves accuracy but also
increases variability in operator judgment. Despite clear guidelines, decisions in shadowed or low-contrast areas,
especially near nunataks or persistent snow, can cause minor inconsistencies. We chose August 2022 imagery to minimize
seasonal snow cover, but regional year-to-year changes in snow and melt can still obscure the true ice margin. This
unquantified temporal uncertainty adds complexity, especially for features near the detection limit.

In Greenland, the typically oblique solar illumination enhances surface contrast in Sentinel-2 imagery highlighting surface
textures, often enabling accurate delineation of land terminating ice. However, mapping remains more challenging for
outlet glaciers, small glaciers and generally those covered with debris, where visual contrast is limited, and glacier
boundaries are less distinct. In contrast, ice caps are generally easier to delineate due to the absence of debris cover. Yet,
the presence of attached snow patches, both seasonal and perennial, introduces considerable uncertainty (Rastner et al.,
2012). On the other hand, in steep, narrow valleys, the same oblique sunlight can cast long shadows that obscure glacier
margins. The shadows blend with adjacent terrain or cause portions of the ice surface to appear visually similar to
surrounding rock or debris-covered slopes. As a result, shadowed areas can lead to underestimation of glacier extent or
misidentification of the true ice boundary. This limitation is especially pronounced in high-relief regions, where accurate
mapping requires optimal solar conditions, multi-temporal imagery, or complementary datasets such as hillshaded digital
elevation models.

The comparison with an independent mapper for the quantification of the omission error highlights both the high
completeness of the PROMICE-2022 nunatak mapping and the subjectivity inherent in manual delineation of small ice-
free features. The low omission error (~1 %) confirms that very few nunataks were overlooked in our mapping, reinforcing
the reliability of the dataset. The large number of nunataks not detected by the independent mapper (n = 1,178) suggests
a more conservative mapping approach or variability in identifying features near the ice margin or in areas of low contrast.
These discrepancies underline the importance of clear mapping protocols and consistent visual criteria, especially when
nunataks are small, partially debris-covered, or located in ambiguous topographic settings. Additionally, it emphasises the
importance of a subsequent quality assessment following both the initial and independent mapping procedures to ensure
confident validation of the final ice margin and nunatak delineation.





### 5.2 Connectivity levels

The delineation of glacier connectivity and the subdivision of ice complexes into distinct units remain challenging and subject to interpretation. As noted by Weidick et al. (1992) separating local GICs from the contiguous GrIS poses a significant cartographic and glaciological challenge. Despite considerable advances since then, a consistent solution applicable across the entire ice sheet has yet to be universally adopted. In the PROMICE-2022 ice mask, we adapted the classification of ice features from Rastner et al. (2012) to the updated GrIS outline, but we did not reanalyse morphological evidence, including flow patterns, topographic divides, and connectivity in the accumulation and ablation zones.

### 5.3 Comparison to other outlines

Most of the small-scale visual differences between the PROMICE-2022 ice mask and earlier GrIS outlines, namely GL50, GIMP, BedMachine, the RGI, and the PROMICE-1987 ice mask, can be attributed to a combination of factors: variations in mapping approaches (e.g., GL50), differences in dataset resolution (e.g., BedMachine), contrasting ice delineation techniques (e.g., GIMP), and temporal changes in the GrIS itself (e.g., PROMICE-1987 ice mask). As a result, direct,

quantitative comparisons of area and perimeter length are not meaningful.

Compared to the GL50 outline, the most recent high-resolution dataset of the GrIS, we applied a different mapping approach by including debris-covered ice as part of the ice sheet. A visual inspection, however, highlights two main areas included in the GL50 outline that we did not consider connected to the PROMICE-2022 ice mask based on our connectivity criteria. One is the local GIC on the Petermann Peninsula in North Greenland, which was previously

connected to the GrIS via Faith Glacier and the floating tongue of Petermann Glacier. In the Sentinel-2 imagery from August 2022, no such connection was observed. However, due to the high interannual variability of marine-terminating glacier tongues, future iterations of the PROMICE ice mask might again include this GIC as part of the GrIS.

The second area is Nuna Dronning Louise in Northeast Greenland, which was previously mapped as a nunatak. In the 2022 imagery, we observed a continuous rift between the termini of the converging glaciers Storstrømmen (flowing from north to south) and L. Bistrup Bræ (flowing from south to north), indicating that Nuna Dronning Louise is no longer a

nunatak but instead lies outside the current GrIS margin. It should be noted that our mapping approach delineates the seaward edge of the floating ice tongues of these two marine-terminating glaciers, rather than the grounding line (Andersen et al., 2025). In contrast, the local GICs on Suess Land and Lyell Land in East Greenland are connected to the GrIS in the PROMICE-2022 ice mask, but not in the GL50 outline. However, most of this area qualifies as CL1, given

that the drainage divides are clearly separated and the GICs do not merge with the GrIS in the ablation zone, as they do not contribute to the GrIS's mass balance.

### 6 Applications and future updates

### 6.1 Uses for the PROMICE-2022 ice mask

This mapping of the PROMICE-2022 ice mask is an important step towards a more comprehensive understanding of

Greenland's cryospheric dynamics and its contribution to global sea-level rise. The PROMICE-2022 ice mask provides a high-quality, expert-based, and scientifically validated GrIS outline to scientific communities, including those working in ice sheet modelling, glacier dynamics, hydrology, remote sensing, and machine learning. The dataset has been thoroughly





validated and expands the availability of training data for automated mapping techniques. The consistently high amount of time required to manually map and validate ice sheet outlines contrasts with the growing availability of high-resolution

satellite data. Furthermore, this accurate and high-resolution outline provides local, regional, and international stakeholders with a base dataset that policy and legislative frameworks addressing ice sheet changes, glacier retreat, and associated impacts on coastal and global systems can be built upon.

Future challenges in ice sheet mapping lie in developing methods to reliably automate this process. The present dataset serves as a resource to train and refine future algorithms. Especially when combined with freely available DEMs and

remote sensing data (e.g., Sentinel-2), automated methods can leverage elevation data and optical imagery to improve mapping accuracy. Additionally, this dataset includes a statistical estimation of mapping errors, providing critical insights into the reliability and completeness of the mapped ice sheet outline.

### 6.2 Community feedback for the PROMICE-2022 ice mask

Delineating the margin of the GrIS involves a degree of interpretation. While we have applied rigorous quality control

throughout the mapping process, we acknowledge that some inaccuracies such as localized misalignments may remain. We therefore welcome feedback from the scientific glaciology community to help improve the PROMICE-2022 Ice Mask dataset. Any issues, suggestions, or corrections can be submitted via the project's GitHub repository (https://github.com/GEUS-Glaciology-and-Climate/PROMICE-ice-mask).

### 6.3 The future of the PROMICE ice mask

With the expected acceleration of ice sheet mass loss due to climate change, a detailed and regularly updated outline of the GrIS is more important than ever. In Greenland, a combination of warming temperatures, increasing melt rates, and dynamic ice flow changes will continue to reshape the GrIS margins. Understanding these changes and their implications is critical for effective climate adaptation strategies. To support this, the PROMICE ice mask will be regularly updated with future inventory years, following the methodology and data sources presented here. We anticipate that upcoming

versions will also include glaciers and ice caps peripheral to the GrIS.

### 7 Data availability

The dataset is openly available on the GEUS Dataverse at https://doi.org/10.22008/FK2/O8CLRE (Luetzenburg et al., 2025) distributed under a CC BY 4.0 license (https://creativecommons.org/licenses/by/4.0/). If the dataset is presented or used to support results of any kind, then we ask that a reference to the dataset be included in publications, along with any

relevant publications from the data production team. If the dataset is crucial to the main findings, we encourage users to reach out to the authorship team as this will likely improve the quality of the work that uses this product. The scripts for processing the data with Sentinel Hub's processing API are provided on the projects GitHub repository https://github.com/GEUS-Glaciology-and-Climate/PROMICE-ice-mask.





## 8 Conclusions

Here, we present the PROMICE-2022 Ice Mask, a high-resolution, manually curated dataset outlining the contiguous ice masses of the Greenland Ice Sheet and the nunataks within its interior as of August 2022. The ice mask is based on a 10 m resolution true-colour Sentinel-2 mosaic and is supported by additional high-resolution satellite imagery and topographic vector data from the Danish Agency for Climate Data. The dataset distinguishes between marine- and land-terminating glacier fronts and classifies nunataks and ice sheet outline with high geometric accuracy (<20 m).

The PROMICE-2022 ice mask provides a consistent, up-to-date baseline for monitoring changes in the extent of the Greenland Ice Sheet. It serves as a resource for glaciological modelling, hydrological studies, and long-term climate change assessments. The dataset is freely available in vector and raster formats and includes metadata on mapping methods and uncertainties. Future updates are planned to include peripheral glaciers.

We invite the scientific community to provide feedback and contribute to further refinements of the dataset via the associated GitHub repository.

### Author contributions

NJK, and RSF conceptualized the study. NJK developed the methodology, with contributions from GL. The investigation was performed by GL, NJK, AKD, TS, KG, and TS. Validation was carried out by GL, NJK, AKD, TS, DF and ESBN. RPM provided resources. Data curation was performed by GL, NJK, and AKD. Formal analysis and visualization were prepared by GL, AKD, and PH. GL and NJK wrote the original draft. All authors contributed to reviewing and editing the manuscript. Supervision and project administration were provided by GL, NJK, and RSF. Funding acquisition was secured by RSF.

### Competing interests

The authors declare that they have no conflict of interest.

### Acknowledgments

We acknowledge input from Ken Mankoff and the *IACS Working Group on the delineation of glaciers, ice sheets, and ice sheet basins* on this data product.

### Financial support

This research has been supported by the Programme for Monitoring of the Greenland Ice Sheet (PROMICE), funded by the Geological Survey of Denmark and Greenland (GEUS) and the Danish Ministry of Climate, Energy and Utilities under the Danish Cooperation for Environment in the Arctic (DANCEA), conducted in collaboration with DTU Space (Technical University of Denmark) and Asiaq Greenland Survey.



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
