# Peer review of "PROMICE-2022 Ice Mask: A high-resolution outline of the Greenland Ice Sheet from August 2022"

_Earth System Science Data, 2025_

## Referee Comment (RC2)

**Review of ESSD-2025-415**

**General comments**

The study by Lutzenburg et al. presents a new dataset with vector outlines for the entire Greenland Ice Sheet that mostly refer to August 2022. To be clear from the start, this is a tremendous achievement and regarding quality a fantastic new dataset that is urgently needed. I only had a quick look at datasets number 6 and 7 in the list and can only hardly imagine how much effort the authors have spent to get it to this point. Indeed, such a dataset is never final and I have also seen a few regions that could be improved (e.g. some debriscovered glacier tongues are incomplete), but as the authors have encouraged feedback to the dataset and provided a link for it, I will not list such issues here. However, I have a few smaller points to the text (see Specific comments) and three larger ones to the datasest:

- (1) The authors provide 12 different dataset versions that might all find their users, but I struggle a bit with the selection for further use and analysis, i.e. which dataset should be used by the community under which conditions? Or, in other words, the product user guide is missing. Section 6.1 is rather general and does not refer to specific datasets. I would like to see a (maybe tabular) list of recommended applications for each dataset so that the users know which one is relevant for them. Which one should be the master dataset?
- (2) My other major point is a missing discussion about the consequences of the temporal development and resulting overlaps and gaps when combining the 2022 ice sheet outlines with other datasets. In Section 5.3 there is a bit of discussion for GL50, but nothing on the RGI. I think this would be important to discuss, as there are now several overlaps with the peripheral glaciers of the RGI and some former CL=2 glaciers (that are not in the RGI) have now disconnected from the ice sheet (changed to CL=1) and are thus also not in this dataset. Hence, a combination with the RGI would result in several missing as well as double counted glaciers. I acknowledge that dataset nr. 7 might have the intention to solve this, but I am not fully sure how to use it in this regard. Should everything with ice divides be removed to avoid overlap? What is with the glaciers that changed from CL=2 to CL=1? Please discuss this.
- (3) My third point is a bit related to points (1) and (2). I support the decision to not change the connectivity levels, but I also see the deviating assignments, often due to a violation of the rules in the RGI dataset (i.e. it includes more peripheral glaciers as CL=1 than it possibly should). However, many of these peripheral ice caps in the RGI are dynamically separated from the ice sheets' outlet glaciers (e.g. do not contribute to their mass flux) and include very small glaciers in steep terrain. These are more the target of glacier models and likely difficult to model with ice sheet models. I do not expect that this issue will be solved here, but I would expect some words about the problem and maybe encourage the related communities to look closer at these topics (e.g. change of CL through time, spatial overlap and resolution, ice divides, modelling limits). This would be very valuable to initiate related activities.

**Specific comments**

L31: glacial retreat: please use glacier retreat when referring to contemporary glaciers.

- L108: A 1 km resolution DEM to orthorectify 10 m satellite images? This might be ok for the interior of the ice sheet but for its steep margins? Please explain what the impacts on geolocation are. Please also explain why neither the ArcticDEM nor the GLO-DEM (from Copernicus) has been used.
- L126: Assuming that manual editing is mostly required for debris-covered glacier parts, how does Douglas-Peucker decrease the amount of work for correcting debris?

- L148: I think 1:25,000 is more the Landsat scale, editing Sentinel-2 data can use 1:10,000.
- L158: I think this study is much more consistent in applying the CL1 definition than the one by Rastner et al. (2012). However, in part also due to shrinking glaciers, this leaves us now with CL2 glaciers in Rastner et al. that have changed to CL1 and are no longer included. On the other hand, many glaciers and icecaps that are now counted as a part of the ice sheet although they were CL1 before. In other words, there is now considerable spatial overlap of the new ice-sheet inventory with the previous local glaciers inventory. As mentioned in the general comments, please discuss this in more detail.
- L170: For a few larger outlet glaciers debris-covered part were partly missed.
- L174: The late in August scenes might suffer from extended regions in shadow which are also very difficult to classify.
- L216: Is the mosaic also available as false colour infrared? This usually provides better contrast for glacier mapping.
- L244: This is the mean error of what? The geolocation? How could there be a systematic shift? Or is it an uncertainty? And what is the impact of the area (e.g. for nunataks)?
- L258/261: Why is the area uncertainty for the entire ice sheet so much smaller than for the nunatak area?
- L246: With so many files being provided, the question arises which one should now be used by the community? Please explain and provide a (maybe purpose dependent) recommendation (see point (1) above).
- L291: I think the uncertainty visualization is fine, but I suggest getting the two figures a bit smaller and side-by-side.
- L310: I think for the northern half of Greenland, snow conditions were much better in August 2019. However, the mapped outlines seem not to be impacted too much by seasonal snow. Is it possible to shortly describe how this has been achieved?
- L320: Has the shadow problem been improved by checking other (high-resolution) images such as provided by Web Map Services (e.g. Google Earth or ESRI World Imagery)?
- L335: Hmmm, as far as I can see, the Rastner et al. study has much more local glaciers/ice caps assigned to CL=1 than this study. I suggest commenting on this.
- L340: In my view, this section should also discuss the differences between the RGI outlines and the new datasets, in particular the regions of overlap and the consequences of now missing glaciers. Are there any solutions to overcome the double counting problem?
- L379: I think the problem of the new dataset is not so much the quality of the delineation but the overlap with existing datasets, changes through time changing the connectivity level (e.g. glaciers might disconnect) and the missing discussion of including peripheral ice caps (and their outlet glaciers) to the ice sheet although they have dynamically little to nothing to do with it and can be much better modelled with glacier models (see above). I acknowledge that this is difficult, subjective and might be inconsistent with the CL1 definition, but many of these now attached ice bodies are difficult to consider in ice sheet models. I acknowledge that the datasets might not be changed now, but at least I would like to see a discussion about these issues I the text (see general comments).

---

## Author Comment (AC1)

**Response to reviewers**

**PROMICE-2022 Ice Mask: A high-resolution outline of the Greenland Ice Sheet from August 2022 (essd-2025-415)**

**AC:** We thank the reviewers for their thorough and constructive assessments of our manuscript. Their comments have significantly improved the clarity, robustness, and overall quality of the work. In this document, we provide detailed responses to all points raised by the reviewers. Reviewer comments appear in black text, and our corresponding responses are provided in green text directly below each comment. All revisions made in the manuscript are indicated in the tracked-changes version submitted shortly after this response. We appreciate the reviewers' time and effort and hope that the revisions satisfactorily address all concerns.

Gregor Luetzenburg

// on behalf of all authors

**RC1**

Luetzenburg et al. describe the production of a new ice mask for the Greenland Ice Sheet. This mask has been manually produced, primarily from 10 m Sentinel-2 RGB mosaic from ~August 2022, supplemented with additional high-res data such as SPOT 6/7. The independent validation and quality control are rigorous, and the dataset clearly represents the culmination of a lot of work. The data is presented in FAIR-aligned open and modern formats (geopackages, etc.) and the existence of a clear GitHub repo for reporting and fixing issues is indicative of the effort the authors have gone to make sure that this dataset will be valuable in the long term. I have very few comments, largely potential opportunities to further reinforce the long-term value of the dataset.

Thank you very much for your positive and thoughtful assessment of our work. We are glad that the rigorous quality control, the use of modern FAIR-aligned formats, and the open issue-tracking workflow were recognised as strengths of the PROMICE-2022 dataset. We also appreciate your suggestions on how to further enhance its long-term value. Below, we address your comments in detail and outline where we have implemented changes or clarifications in the manuscript and documentation.

**DATASET COMMENTS**

My initial reaction on seeing the total size of the dataset was that it was surprisingly large (the best part of a GB) relative to my expectation. This is due to the fact that the authors provide a number of options and variations on the dataset (polygons, lines, rasters, with/without nunataks, etc etc). On balance, I think that the benefit of these options outweighs the downside of having an inflated total dataset size - I have definitely been frustrated in the past by e.g. needing a vector dataset when a raster is available (and vice

versa) and having to convert between these types. However, I might recommend that the 'detailed description' on the Dataverse also includes the brief list of file descriptions that is available on `00-README-PROMICE-2022-IceMask.md`. This would allow users to quickly identify and download only the most relevant file for their use case, without needing to download the separate markdown file.

Thank you for this helpful suggestion. We have now added the full list of file descriptions from the 00-README-PROMICE-2022-IceMask.md directly to the "Detailed Description" section on Dataverse. This ensures that users can immediately see the purpose of each file and download only the datasets relevant to their needs, without first retrieving the separate README file.

One thing I have noticed is that the dataset itself does not currently include the semantic version number, although I am aware that the Dataverse file does (currently v2.1). The very useful setup of the GitHub would imply that, hopefully, a number of future version updates are to come. However, after downloading the dataset, there is nothing that I can see - either in the README.md, nor any file metadata, that would indicate the version number. I can easily imagine a situation where the dataset is downloaded by a user for a project, a year passes, and then as a paper is being written up the user has no idea what version of the dataset is used – even as a number of updates have been made to the main dataset. The simple/idiot-proof thing to do would be to include the version number in all the individual filenames (e.g. `01-PROMICE-2022-v2.1-IceMask-line.gpkg`), although maybe the authors have a better suggestion.

Thank you for pointing this out, it is an excellent suggestion. We have now added the version number directly to all filenames, e.g. 01-PROMICE-2022-IceMask-line-v3.gpkg. Whenever data files are re-uploaded, Dataverse automatically increments the major version (e.g., from v2 to v3). In contrast, metadata-only edits result in a minor version update (e.g., v3.1). Because we re-uploaded the files after adding the version number to each filename, Dataverse correctly created a new major version (v3).

Why was the choice made to not provide a higher-resolution raster? I understand that a 150 m option aligned with BedMachine will be useful for modelling purposes, but a high-res version (10 m) would be useful for masking satellite data at a higher resolution, and would compete with the GrIMP mask (15 m resolution). Although I understand that the file size may be the main limitation, I wonder whether a properly zstd-compressed boolean dataset would be smaller than suggested by a proportional scaling of a 4.4 MB 150 m resolution dataset (which would imply a 10 m dataset approaching a gigabyte). Perhaps it could be made available in a separate repository for those interested.

Thank you for the helpful comment. In addition to the 150 m raster aligned with BedMachine, we now also provide 10 m sparse rasters for (i) the ice-sheet outline, (ii) the nunataks, and (iii) a combined outline + nunatak mask. These rasters include only the margin cells rather than the full ice-sheet interior, which keeps file sizes manageable

while still enabling a high-resolution at 10 m. Providing a full 10 m interior raster would indeed approach hundreds of gigabyte scale even with efficient compression, whereas the sparse representation allows us to maintain high spatial fidelity along the margin without the storage burden of a complete high-resolution dataset. Users who need a filled 10 m mask can readily generate it by applying standard NoData-filling tools in GDAL/QGIS, as the sparse rasters form a complete closed boundary.

The basin polygons (files `08` and `09`) do not currently include the `NAME` column from the original Mouginot dataset. Whilst the column names aren't perfect (lots of `NW_NONAME1` style fudges), it would help to provide some context to the polygons, as well as providing an appropriate way to `join` this dataset with others also based on the Mouginot basins.

Thank you for pointing this out. We have now added the NAME column from the original Mouginot dataset to both basin polygon files (08 and 09). This ensures full compatibility with other datasets using the same basin scheme and provides clearer context for each polygon.

One interesting consequence of the choices made in producing the Cl1 dataset (file `07`) and the basins polygon (file `08` and `09`) is that the CL1 polygon divides the peripheral ice caps but not the ice cap, whilst the basins polygon divides the ice sheet but not the peripheral ice caps. It's clear why this was done - one is consistent with Rastner et al (2012), whilst the other is consistent with the Mouginot et al (2019) dataset (L158-161). It does provide an interesting inconsistency: it might be frustrating if one were interested in basins/ice divides for both Greenland and peripheral ice caps. I'm not sure if this is common enough to deserve a separate 'combined' dataset, but perhaps just something worth thinking about.

Thank you for this observation. To address this, we added the connectivity levels from file 07 to the basin delineations in files 08 and 09 as a separate column. Each basin is assigned either CL1 or ice mask based on the largest area of overlap between the CL1 polygons and the basin polygons. This ensures consistency across datasets while maintaining compatibility with both the Rastner et al. (2012) and Mouginot et al. (2019) frameworks. Please see more details below in the response to comment 2 and 3 from RC2.

It is a missed opportunity, given the effort in producing and describing the August 2022 Sentinel-2 mosaic, that it is not made available for users. The ITS_LIVE project makes available a similar mosaic (for 2019) as Cloud-optimised Geotiffs (CoGs) and a GUI-GIS-friendly .vrt file on their AWS bucket (https://its-live-data.s3.amazonaws.com/index.html#rgb_mosaics/GRE2/), although it is (to my knowledge) not well documented nor advertised. Having an equivalent dataset that is appropriately described and citeable would be useful to the community for providing a

FAIR and citable Greenlandic mosaic for visualization exercises. This could be extremely low-hanging fruit in the context of this paper.

On the topic of CoGs, I wasn't able to validate whether data is available in a cloud-optimized geospatial formats to allow users to download the required data directly within their code without having to download the full dataset, as you can currently do with e.g. ArcticDEM/REMA and ITS_LIVE data. I couldn't manage to test this with the Dataverse files as I'm not sure it's possible to get direct links to the datasets. I'm not sure that the geopackage format is cloud-optimised anyway (If I remember correctly vector datasets probably require a geoparquet or geojson to be cloud-friendly). If it is possible, perhaps it might be possible to include a Jupyter Notebook on the GitHub to show users how.

Thank you for these helpful comments regarding the accessibility and cloud-friendly availability of the August 2022 Sentinel-2 mosaic. We have now processed all Sentinel-2 raster files into Cloud-Optimized GeoTIFFs (COGs) and made them available through the Dataverse repository. To ensure usability and manageable file sizes, the dataset is provided as eight regional mosaics. These mosaics can be accessed directly from Dataverse and used in GIS software or through standard HTTP range requests supported by the COG format. At this stage, we do not plan to provide a Jupyter Notebook, but the COG structure ensures that users can efficiently stream and subset the data within their own workflows without downloading the full mosaics. We hope that this improves the FAIRness, accessibility, and practical usability of the dataset for the community.

**MINOR COMMENTS**

L50-54 - long-running sentence, could do with being split!

We split the sentence in two.

L136 - Debris is referred to as challenging (L320, etc), but I think it is slightly under-discussed within the methods as to any particular guidance provided to the delineators when difficult choices were made. How was the SPOT6/7 data helpful in determining debris? Was it a case of visualizing surface roughness, shadows, etc?

We added the following to the method section: We assumed that the basal thermal state of Greenland's outlet glaciers is predominantly thawed and therefore they occupy the full width of the U-shaped valleys that constrain their flow, particularly along the steep fjord systems of eastern Greenland (Macgregor et al., 2022, https://doi.org/10.5194/tc-16-3033-2022). In cases where Sentinel-2 imagery did not allow an unambiguous distinction between glacier ice, supraglacial debris, and shadow, most commonly in narrow valleys, we supplemented the optical images interpretation with ArcticDEM data. Hillshade visualizations and surface-roughness metrics derived from ArcticDEM were used to identify the transition between glacier surfaces (including debris-covered ice) and the topography of adjacent valley walls. This combined approach allowed us to consistently map glacier boundaries in areas where optical data alone were insufficient.

L143-150 - Perhaps an additional useful practical analogue in this paragraph would also be the approximate distance between polygon vertices at dense and coarsely mapped sectors of the ice sheet?

We describe the simplification and densification in section 3.3: The vector lines representing the margin and each individual nunatak were then dissolved and simplified by distance, using a tolerance of 5 m, equivalent to half a pixel (Douglas and Peucker, 1973). Mean error of the simplification is 1.3 m. Subsequently, the simplified lines were densified, ensuring a maximum spacing of 100 m between vertices.

L256 - In various places around the paper (I first started picking up on this in this paragraph), the ice mask vector is stated to represent 'glacierized areas' of the Greenland Ice Sheet. I am unsure whether there is a consensus that a 'glacierized area' would include floating ice, as your dataset does. Perhaps it is worth stating explicitly in the abstract and top-level README.md files that the dataset includes floating ice?

We clarified this both in the abstract and in the results section.

L340/Section 5.3 It is a shame that this section is not accompanied by a figure that could visualize the various alternative masks in a few locations.

We agree that comparing the various alternative ice masks would be very interesting. However, we feel that such a comparison would go beyond the scope of this data description paper. The IACS Working Group on the delineation of glaciers, ice sheets, and ice sheet basins is currently preparing a report that compares the existing delineations (see: https://iacs-cryo.github.io/Delineation-WG/deliverable1/).

L393-395 - Perhaps also recommend a citation of this publication as well as the dataset citation, as is consistent with your GitHub terms of use as well as generally aligned with other dataset citation requests?

We clarified now reading: If the dataset is presented or used to support results of any kind, then we ask that a reference to the dataset be included in publications, along with this data description paper and any relevant publications from the data production team.

Figure 6b - Is this Venn diagram meant to have areas proportional to the data counts? If so, like with pie charts, this is probably hard for humans to meaningfully ingest and could probably be better represented with e.g. a stacked bar chart with a single (horizontal) bar.

Thank you for this constructive suggestion. The Venn diagram is indeed intended to represent areas proportional to the underlying data counts. While a horizontal stacked bar chart is well suited for comparing total counts, it cannot directly represent the overlap between the independent QC mapping and the PROMICE-2022 ice mask, which is the key piece of information we aim to communicate in this panel. To aid interpretation, we have explicitly added the numerical values for each category to the figure so that readers

who are less familiar with Venn diagrams can still readily understand the relationship between the two datasets.

**RC2 – Frank Paul**

**General comments**

The study by Lutzenburg et al. presents a new dataset with vector outlines for the entire Greenland Ice Sheet that mostly refer to August 2022. To be clear from the start, this is a tremendous achievement and regarding quality a fantastic new dataset that is urgently needed. I only had a quick look at datasets number 6 and 7 in the list and can only hardly imagine how much effort the authors have spent to get it to this point. Indeed, such a dataset is never final and I have also seen a few regions that could be improved (e.g. some debris-covered glacier tongues are incomplete), but as the authors have encouraged feedback to the dataset and provided a link for it, I will not list such issues here. However, I have a few smaller points to the text (see Specific comments) and three larger ones to the datasest:

Thank you very much for your thoughtful and constructive feedback, and for recognising the effort that went into producing the PROMICE-2022 dataset. We greatly appreciate your positive assessment of the overall quality and the usefulness of the product. As you noted, such a dataset can never be entirely final, and we welcome all suggestions for improvement. We are grateful that you will share specific feedback through the provided reporting link. Below, we respond to your three main points and the specific comments you raised.

(1) The authors provide 12 different dataset versions that might all find their users, but I struggle a bit with the selection for further use and analysis, i.e. which dataset should be used by the community under which conditions? Or, in other words, the product user guide is missing. Section 6.1 is rather general and does not refer to specific datasets. I would like to see a (maybe tabular) list of recommended applications for each dataset so that the users know which one is relevant for them. Which one should be the master dataset?

Thank you very much for this helpful comment. We agree that clear guidance is essential given the number of datasets we provide. In response, we have now added a Product User Guide (see new Table 3), which summarizes recommended use cases for each dataset and identifies the intended "master" datasets for most applications. The guide includes a tabular overview that links each product to typical use cases, analysis workflows, and recommended situations. It also clarifies how the different line, polygon, and raster products relate to each other and how they should be selected depending on the user's needs (e.g., outline delineation, masking satellite imagery, basin-based modelling, glacier connectivity analysis, etc.).

(2) My other major point is a missing discussion about the consequences of the temporal development and resulting overlaps and gaps when combining the 2022 ice sheet outlines with other datasets. In Section 5.3 there is a bit of discussion for GL50, but nothing on the RGI. I think this would be important to discuss, as there are now several overlaps with the peripheral glaciers of the RGI and some former CL=2 glaciers (that are not in the RGI) have now disconnected from the ice sheet (changed to CL=1) and are thus also not in this dataset. Hence, a combination with the RGI would result in several missing as well as double counted glaciers. I acknowledge that dataset nr. 7 might have the intention to solve this, but I am not fully sure how to use it in this regard. Should everything with ice divides be removed to avoid overlap? What is with the glaciers that changed from CL=2 to CL=1? Please discuss this.

We agree that a clearer discussion of the consequences of combining the 2022 PROMICE ice-sheet outlines with other temporal datasets is needed. We have therefore expanded Section 5.2 to include a dedicated paragraph discussing:

- overlaps between the PROMICE-2022 ice mask and the RGI v6.0 outlines
- issues arising from temporal inconsistencies in glacier connectivity
- the effect of glaciers that transitioned from CL2 to CL1
- the risk of double counting or missing glaciers when combining datasets
- guidance for users on how to work with file 07 (CL1 polygons) and files 08/09 (basins) – Section 3.3

We now clarify that CL1 polygons provide higher spatial detail than the Mouginot basins, and we intersected the basins with the CL1 glaciers. This information is now included in file 09, allowing users to work with basin-based delineations that incorporate the smaller CL1 catchments.

(3) My third point is a bit related to points (1) and (2). I support the decision to not change the connectivity levels, but I also see the deviating assignments, often due to a violation of the rules in the RGI dataset (i.e. it includes more peripheral glaciers as CL=1 than it possibly should). However, many of these peripheral ice caps in the RGI are dynamically separated from the ice sheets' outlet glaciers (e.g. do not contribute to their mass flux) and include very small glaciers in steep terrain. These are more the target of glacier models and likely difficult to model with ice sheet models. I do not expect that this issue will be solved here, but I would expect some words about the problem and maybe encourage the related communities to look closer at these topics (e.g. change of CL through time, spatial overlap and resolution, ice divides, modelling limits). This would be very valuable to initiate related activities.

We agree that the broader implications of changing connectivity levels, spatial overlaps between datasets, and modelling challenges deserve explicit discussion. We have added a new paragraph in Section 5.2 addressing:

- why we preserved the CL definitions of Rastner et al. (2012)
- how temporal changes in glacier extent cause CL transitions (e.g. CL2 → CL1)
- how this creates spatial inconsistencies when comparing with older datasets
- the need for community-wide work on CL definitions, temporal evolution, and integration across inventories

This paragraph also directly addresses RGI-specific issues raised here and in comment 2 and explains that our dataset intentionally avoids redefining ice divides on peripheral ice caps even where separation is dynamically reasonable.

**Specific comments**

L31: glacial retreat: please use glacier retreat when referring to contemporary glaciers.

Ok, we changed glacial to glacier.

L108: A 1 km resolution DEM to orthorectify 10 m satellite images? This might be ok for the interior of the ice sheet but for its steep margins? Please explain what the impacts on geolocation are. Please also explain why neither the ArcticDEM nor the GLO-DEM (from Copernicus) has been used.

For this specific data product, KDS states that they orthorectified the Sentinel-2 mosaics using the GLOBE (1 km) or SRTM (30 m) DEMs. ESA now typically uses the Copernicus DEM (30 m resolution) for orthorectification in most cases. In our study, the Sentinel-2 mosaic provided by KDS was of secondary importance, as we used the August 2022 Sentinel-2 mosaic for the vast majority of our mapping.

To account for any resulting geolocation uncertainty in the PROMICE-2022 ice mask, we independently mapped the geometric uncertainty. This allows users to assess the reliability of mapped boundaries, including in areas where steep topography may introduce additional positional error.

L126: Assuming that manual editing is mostly required for debris-covered glacier parts, how does Douglas-Peucker decrease the amount of work for correcting debris?

When editing the existing outline, each vertex must be adjusted manually. By reducing the number of vertices, we can significantly decrease the amount of manual editing required.

L148: I think 1:25,000 is more the Landsat scale, editing Sentinel-2 data can use 1:10,000.

The 1:20,000 scale for Sentinel-2 imagery is based on standard cartographic guidelines (Tobler, 1987) and the 10 m image resolution, which corresponds to a minimum mapping feature (MMF) of approximately 1 mm on the map. While finer scales such as 1:10,000 could be used for digitizing, 1:20,000 ensures that features are represented accurately without over-interpreting the data. More importantly, for the delineation of the GrIS

margin and nunataks, we used a variable mapping scale adapted to the level of detail required in different regions.

L158: I think this study is much more consistent in applying the CL1 definition than the one by Rastner et al. (2012). However, in part also due to shrinking glaciers, this leaves us now with CL2 glaciers in Rastner et al. that have changed to CL1 and are no longer included. On the other hand, many glaciers and icecaps that are now counted as a part of the ice sheet although they were CL1 before. In other words, there is now considerable spatial overlap of the new ice-sheet inventory with the previous local glaciers inventory. As mentioned in the general comments, please discuss this in more detail.

We expanded the discussion on CL changes over time and clarified the overlap between the PROMICE-2022 ice mask and earlier inventories. This section now explicitly states that some glaciers previously mapped as CL1 are now merged with the ice-sheet margin due to retreat, whereas some glaciers previously mapped as CL2 have become CL1 and are therefore not included in the ice-sheet-only products.

L170: For a few larger outlet glaciers debris-covered part were partly missed.

We kindly ask to submit any suggested changes to the outline via https://github.com/GEUS-Glaciology-and-Climate/PROMICE-ice-mask.

L174: The late in August scenes might suffer from extended regions in shadow which are also very difficult to classify.

We discuss oblique solar illumination and resulting shadows in section 5.1

L216: Is the mosaic also available as false colour infrared? This usually provides better contrast for glacier mapping.

Unfortunately, the mosaic is only available in RGB.

L244: This is the mean error of what? The geolocation? How could there be a systematic shift? Or is it an uncertainty? And what is the impact of the area (e.g. for nunataks)?

This only refers to the error introduced by the simplification applied to the outline after manual mapping. Geometric uncertainty and operator bias are assessed after the simplification and are described independently in Sections 3.2.1 and 3.2.2.

L258/261: Why is the area uncertainty for the entire ice sheet so much smaller than for the nunatak area?

The difference in relative area uncertainty is due to the geometry of the mapped features. The nunataks have a total outline length of 83,676 km and a projected area of 17,916 km$^2$, resulting in an area uncertainty of ±1,674 km$^2$. In contrast, the Greenland Ice Sheet has a perimeter of 53,060 km and a total area of 1,725,648 km$^2$, with an uncertainty of ±1,061 km$^2$. Although the nunataks occupy a much smaller total area, they have a much higher

perimeter-to-area ratio. As a result, the same absolute boundary uncertainty (19.5 m) produces a proportionally larger area uncertainty for the nunataks. The Ice Sheet outline, being larger, is less sensitive to the same margin uncertainty, leading to a smaller relative contribution to area uncertainty.

L246: With so many files being provided, the question arises which one should now be used by the community? Please explain and provide a (maybe purpose dependent) recommendation (see point (1) above).

Please see you reply to your point 1 above.

L291: I think the uncertainty visualization is fine, but I suggest getting the two figures a bit smaller and side-by-side.

Thank you for your suggestion. We chose to keep the two panels on top of each other.

L310: I think for the northern half of Greenland, snow conditions were much better in August 2019. However, the mapped outlines seem not to be impacted too much by seasonal snow. Is it possible to shortly describe how this has been achieved?

We deliberately chose images from August to minimize seasonal snow cover. While other years, such as 2019, may have had even less snow in the northern half of Greenland due to high interannual variability, our goal was to use images from a single year and a single month to ensure consistency within the ice mask. Using multiple years could introduce inconsistencies in the mapped outlines. In future iterations, we plan to include additional years, and 2019 could be considered for the next update of the PROMICE ice mask.

L320: Has the shadow problem been improved by checking other (high-resolution) images such as provided by Web Map Services (e.g. Google Earth or ESRI World Imagery)?

Yes, in some areas using the Sentinel-2 and SPOT6/7 images provided by KDS as well as ArcticDEM has improved the shadow problem.

L335: Hmmm, as far as I can see, the Rastner et al. study has much more local glaciers/ice caps assigned to CL=1 than this study. I suggest commenting on this.

We added a short statement noting that Rastner et al. (2012) assign more local glaciers/ice caps to CL1 than PROMICE-2022, and we discuss why this occurs.

L340: In my view, this section should also discuss the differences between the RGI outlines and the new datasets, in particular the regions of overlap and the consequences of now missing glaciers. Are there any solutions to overcome the double counting problem?

We inserted a paragraph comparing PROMICE-2022 with RGI, discussing overlaps, missing glaciers, and how file 09 (basins and CL1 intersection) can help reduce double counting when combining datasets.

L379: I think the problem of the new dataset is not so much the quality of the delineation but the overlap with existing datasets, changes through time changing the connectivity level (e.g. glaciers might disconnect) and the missing discussion of including peripheral ice caps (and their outlet glaciers) to the ice sheet although they have dynamically little to nothing to do with it and can be much better modelled with glacier models (see above). I acknowledge that this is difficult, subjective and might be inconsistent with the CL1 definition, but many of these now attached ice bodies are difficult to consider in ice sheet models. I acknowledge that the datasets might not be changed now, but at least I would like to see a discussion about these issues I the text (see general comments).

We added text clarifying that the major challenges are not delineation quality but rather:

- temporal changes in connectivity

- spatial overlap with legacy datasets

- how dynamically distinct peripheral ice caps are treated in ice-sheet vs. glacier modelling

We emphasise that although we do not redefine CL assignments, we acknowledge modelling and inventory-integration challenges.